# Analysis of the Control System for a Soft Starter of an Induction Motor Based on a Multi-Zone AC Voltage Converter

**Evgeniy Kosykh** [1,2,*], **Aleksey Udovichenko** [1], **Nikolay Lopatkin** [3], **Gennadiy Zinoviev** [1], **Evgeniy Grishanov** [1,2] and **Regina Sarakhanova** [1,2]

1 Department of Electronics and Electrical Engineering, Novosibirsk State Technical University, 630073 Novosibirsk, Russia
2 Institute of Power Electronics, Novosibirsk State Technical University, 630073 Novosibirsk, Russia
3 Department of Mathematics, Physics, Informatics, Shukshin Altai State Humanities Pedagogical University, 659333 Biysk, Russia
* Correspondence: kosykh@corp.nstu.ru

**Abstract:** The development trends of the modern world of power electronics dictate the requirements for the use of AC voltage converters as soft starters for induction motors. A direct connection of the motors to the mains voltage negatively affects both the motor itself and the mains system as a whole due to high starting currents values, which entail, as a rule, more frequent accidents and shorten the drive system service life. Modern methods of motor acceleration are implemented in practice by means of frequency converters, which require the presence of both a rectifier and an inverter in the structure of the device. The paper presents a study of the control system of a multi-zone AC voltage single-stage regulator. The use of capacitive voltage dividers will also compensate for the consumed reactive power. The article analyzes the features of modern soft starters, describes the circuit design, presents a mathematical calculation by the differential equations algebraization method, a performed simulation modeling in Matlab/Simulink, and also an assembled experimental stand. Particular attention is paid to the definition of the multizonality concept of the proposed converter and the analysis of the control method. The developed algorithm of the double-loop automatic control system will minimize the influence of induction motors on the mains voltage and thus improve electromagnetic compatibility.

**Keywords:** AC voltage converter; multi-zone regulation; soft starter; induction motor; reactive power compensation; improved electromagnetic compatibility

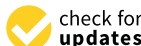



## 1. Introduction

In recent decades, induction motors (IMs) have been the most common load in electrical networks, in particular in network pumps, air pumps, smoke exhausters, fans, and fuel supply equipment. Induction motors are directly affected by all the problems associated with the deviation of the mains voltage from the standard parameters. AC voltage regulators (RAVs) help to minimize the impact of these problems. At the strategic facilities of the residential sector, related to the municipal infrastructure of the city, an important role is played by power converters required for personal needs, providing the city with heat, electricity and water.

Moreover, at present, about 50–80% of the electricity consumed in the world is accounted for AC electric drives of various power ranges. Most often, the type of electric motor drive is unregulated. With all this, induction motors connected directly to the supply voltage network are directly affected by all the problems associated with the low quality of the mains voltage. Low voltage quality leads to a deviation of the consumed current from a sinusoidal form. At present, the structural complication of power supply systems is caused by an increase in the number of electricity consumers. An increase in the mains voltage causes increased consumption of current and reactive power, which entails overheating of

the motors, an increase in repair and maintenance costs, and a reduction in its service life. At the same time, regulation of the voltage on the motor in the steady state of its operation after reaching the rated speed and the ability to control the starting current during soft start will increase its service life.

The widespread use of power electronics converters is the reason for this branch of engineering to become a priority direction in the development of the global electric power industry. Large values of electrical energy losses are typical for distribution networks in the power range from 30 kW to 250 kW. Thus, the requirements for the functionality of such converters are formulated: regulation of the output AC voltage amplitude, reactive power compensation and stabilization of the load voltage when the supply voltage deviates from the standard parameters. One of the most used in practice converters with the function of smooth voltage increase is either the well-known classical thyristor RAV with phase control [1–3] or its circuit analogs with the function of soft power reduction at start-up [4–7]. However, it cannot be used as a solution to these problems due to the low quality of the currents at the input and output of the converter. The reasons are a low frequency (LF) switching and a poor power factor (PF). Increasing the switching frequency range of transistor switches will eliminate these shortcomings.

The most traditional AC voltage regulator structure is the two-stage one, containing a rectifier unit and an inverter unit. The modern multi-level voltage source inverters (MLVSIs) have numerous advantages compared to the two-level ones, including a higher level of generated voltage at a low voltage rating of the semiconductor switches. The existing and emerging baseline MLVSI control methods offer not only a high quality of the output power (generated voltage and current) from the point of view of the harmonics content but also keep a low level of switching losses, which could eliminate or mitigate the problem of a common-mode voltage etc.; see, for instance, [8–10]. Unfortunately, despite the above-mentioned MLVSI features, the overall RAV efficiency remains not of very high value, mostly due to the double energy conversion. In addition, a high number of the circuits' elements, a complicated control, weight and size indicators and a high cost limit the attractiveness of such RAV's structures.

The structure of the AC voltage transformer regulator is relevant for systems with the ability to switch the transformation windings. The use of a transformer-type converter to control the output voltage level in practice is implemented very limitedly. The use of a transformerless type is advantageous when it is necessary to reduce the weight and size indicators and when the range of output voltage regulation is zero. The main contribution to the efficiency calculation of the use of converters is the consideration of dynamic losses during the switching of power switches. Reducing the number of switches in the structure of the regulator will have a positive effect on energy performance.

A direct connection of induction motors to the mains voltage has a negative effect both on the motor itself and the mains system as a whole due to the large values of starting currents and, as a rule, an increase in the frequency of accidents and a reduction in the service life. The existing soft start methods impose requirements for the use of either of the systems with a transformer structure unit or with thyristor switches. The first analog has a negative effect on the weight and size of the converter and has a mechanical type of switching. The second analog has a negative effect on the shape of the consumed current due to LF switching, having a poor spectral composition, and therefore worsening electromagnetic compatibility (EMC) with the power supply network.

The characteristics of the motor's direct connection to the network can be considered huge starting currents of the stator, reaching up to seven times the nominal value. The first thing that will feel harm from such values of starting currents will be the insulation of the stator windings. Wear of the insulation will lead to overheating and, accordingly, to premature wear, which will adversely affect the fastening of the winding due to the occurrence of electrodynamic forces. Starting current surges affect not only the motor itself but also the power supply network. A negative impact on the power network should be understood as a decrease in voltage throughout the network, which negatively affects the

operation of other consumer devices connected to it. Currently, special attention is paid to emergency cases, as well as their analysis [11,12] and diagnostics [13,14], with a discussion of the economic aspects of their application [15].

It is most important to eliminate this in power systems, long lines and when connecting redundant power supplies. A high-power induction motor draws very high currents with a poor PF when starting from the power supply. Normally, soft start is used to avoid this problem and achieve a soft start of high-power induction motors [16,17], but it needs to be optimized [18].

The purpose of soft starters is the ability to control processes at all stages of starting an electric motor: not only smoothly accelerate but also smoothly slow down [19,20]. The payment for a two-fold and three-fold decrease in the starting current in modern common thyristor soft starters for AC motors can be called a significant deterioration in the current shape under load and the appearance of an excess reactive component. Low values of the motor PF lead to premature wear of the IM and increase maintenance costs due to electrical and thermal overloads [21].

Trends in the development of modern control systems (CS) for power electronics converters dictate the requirements for the complexity of automation processes in soft starters for induction motors. In the field of predictive algorithms, CSs can be implemented both closed-loop using current, rotation, and magnetic flux sensors [22–24] and open-loop without using feedback sensors [25].

Currently, the development and research of new designs in the design of turbines for torque minimization include calculations using genetic algorithms and surrogate models based on artificial neural networks (ANNs) and cloud computing [26–31]. A similar approach, but for working with soft start systems, can be applied at the next stages of the study. Along with control algorithms for soft starters of IMs based on an ANN [32–35], artificial intelligence systems [36], observers [37] and fuzzy control [38] are also used.

For the optimal and rational selection of an AC converter, the preferred aspects are:

- transistor power switches due to the high switching frequency, and hence the improved shape of the consumed current when using the Fourier transform (FT) in the decomposition of the spectral composition;
- transformerless structure due to small weight and size indicators;
- voltage conversion factor (VCF) in the range of 0 to 1;
- double-loop automatic CS on start current and motor speed;
- linear control characteristics for simplified development of a microprocessor control system (MPCS);
- multi-level implementation for connecting high-power motors.

Thus, this article is devoted to the analysis of the control system of a multi-zone RAV [39,40] as a soft starter for an IM [41,42].

In this article, we are considering how to control a multi-zone AC voltage converter as a soft starter for induction motors. First, we describe the circuit design of this converter and also reflect on the optimal choice of the number of zones based on energy characteristics in Section 2. Then we carry out a mathematical calculation using the differential equations algebraization method (DEA) and also show a visual principle of the regulator using the system oscillograms control, control pulses and spectral composition in Section 3. Section 4 presents the results of simulation modeling of an induction motor with a direct connection to the network and using a soft starter. Control methods are presented using a relay controller, using a single-loop system of automatic control by starting current and using a double-loop system by motor speed. Cases with a nominally loaded motor shaft are also considered. The experimental stand and oscillograms of the voltage and current of the active load and its spectral composition are presented in Section 5. The analysis of the above in Section 6 allows us to formulate a discussion of the results obtained and note the key points of emphasis. Section 6 also presents interpretations about the conclusions of the optimal converter used to solve the problems of large starting currents and thoughts

on future directions for continuing research. All patents received by the authors of the research are presented in Section 7.

## 2. AC Voltage Converter

A single-phase electrical circuit of a three-zone converter in Figure 1 structurally consists of the following elements: sinusoidal mains voltage (E), capacitor voltage dividers ($C_1$, $C_2$ and $C_3$), active load ($R_{Load}$) and semiconductor elements ($S_1$–$S_6$), which together allow with a MPCS to set and implement the required level of output voltage.

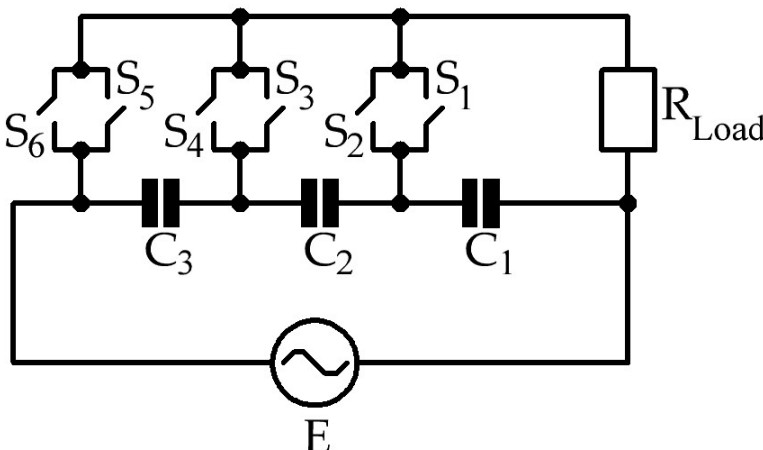

**Figure 1.** Electrical diagram of single-phase converter.

The proposed converter can be implemented in both single-phase and three-phase versions in Figure 2.

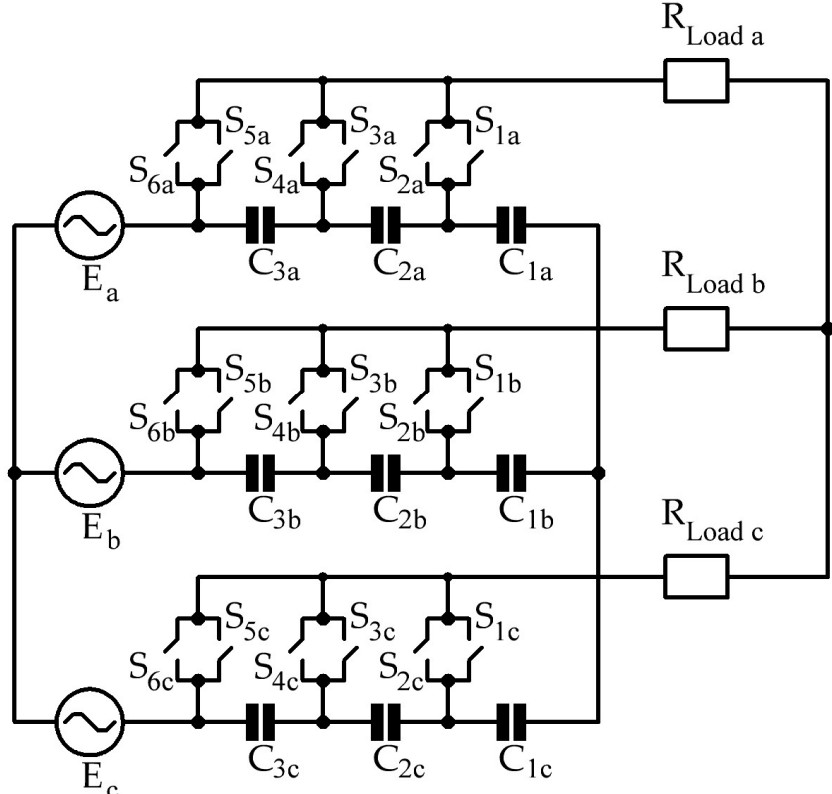

**Figure 2.** Electrical diagram of three-phase converter.

A combination of bi-directional transistors with diodes is used as a switch with bidirectional conductivity and the ability not to short-circuit the capacitor during switching in Figure 3.

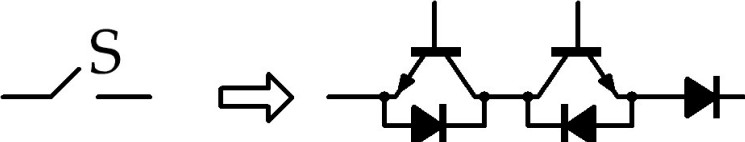

**Figure 3.** Switch equivalent circuit.

For a VCF, the display of control characteristics can be displayed in relative form by dividing the value of the output voltage by the value of the input. The multi-zone structure of the converter CS can be presented in the form of combinations. Relative control characteristics for four different converters are presented in Figure 4.

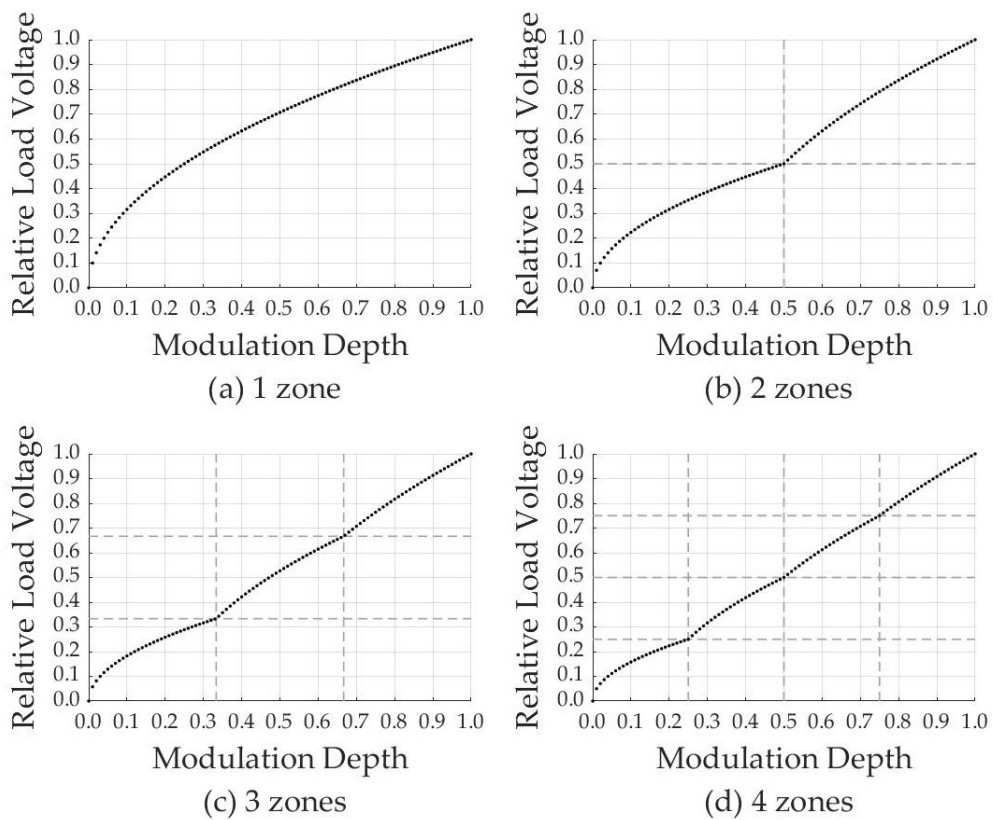

**Figure 4.** Control characteristics (dependences of relative load voltage on modulation depth).

All characteristics are monotonically increasing dependences on the modulation depth and also have key points (1/2 for two-zone, 1/3 and 2/3 for three-zone, 1/4, 2/4 and 3/4 for four-zone). The first aspect is explained by increasing the output voltage of the converter using a pulse width modulation (PWM) control system. The second aspect is characterized by the opening of the zones and the connection of the divided mains voltage between the capacitors to the load. An increase in the number of transducer zones makes it possible to obtain a more linear nature of the control characteristics due to an increase in the number of key points and, therefore, simplifies the development of an MPCS. However, an increase in the number of semiconductor switches entails an increase in power losses for switching with a decrease in efficiency.

With the full opening of each zone (at key points), the shape of the output voltage corresponds to an ideal sinusoidal signal, which means that when decomposed into a

spectrum, harmonics other than the main one do not have. What does the zero contribution of higher harmonics mean in the calculation of the total harmonic distortion (THD) in Figure 5. Since the modulation depth affects the output voltage shape, when the signal is decomposed into a spectral composition, the quality of the load voltage will also change. An increase in the number of zones will also have a positive effect on the quality of the signal since there will be more key points, and the higher harmonics will make a smaller contribution. The multi-zone structure of the AC voltage converter also allows for increasing the voltage range of use and connection of consumer devices to high-voltage networks without harm to semiconductor switches. In the following, the structure of a multi-zone RAV with three zones will be considered.

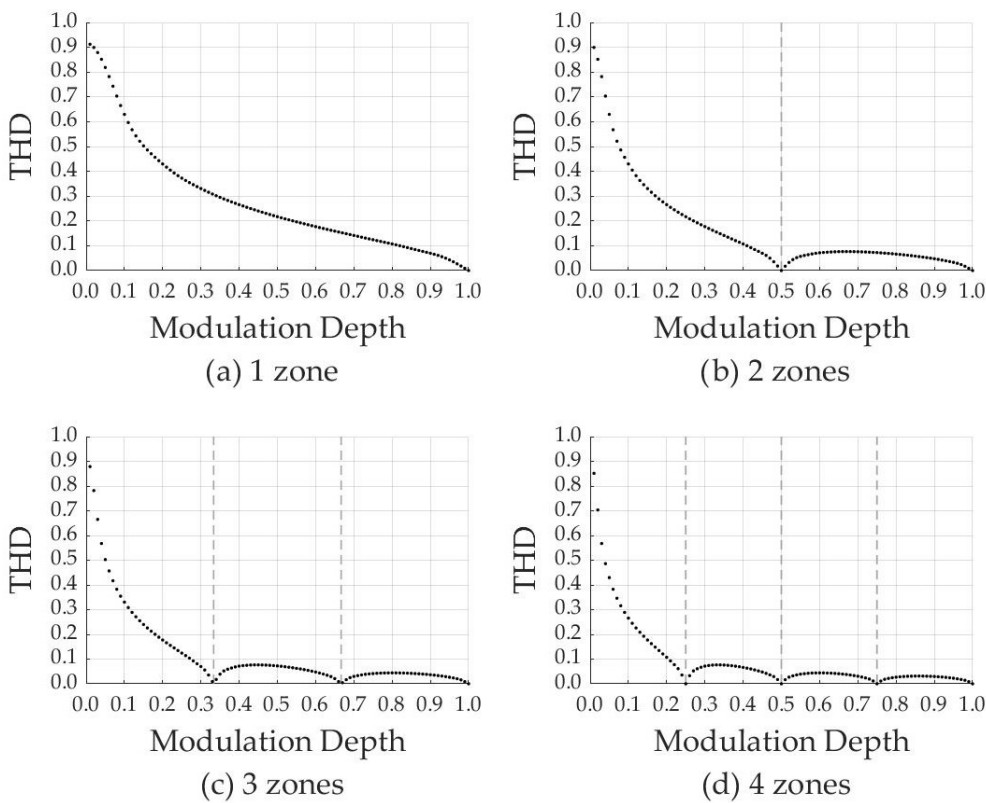

**Figure 5.** THD Characteristics (dependences of THD on modulation depth).

### 3. Mathematical Calculation

For a three-zone RAV, a detailed mathematical calculation is presented by the differential equations algebraization method. Modification of the converter circuit changes depending on the modulation depth *M*. This method will allow finding the effective values of the load current fundamental LF harmonic. To perform this, it is necessary to draw up Kirchhoff's laws to describe the electrical circuit of the converter in four cases.

Since there are exactly four reactive elements in the scheme under consideration ($C_1$, $C_2$, $C_3$ and L), it is necessary to compose a system of equations of the fourth order. It will be painted precisely those values whose derivatives can be expressed through other parameters of the converter, namely three capacitance currents ($i_{C1}$, $i_{C2}$ and $i_{C3}$) and load voltage ($u_2$). For clarity, Table 1 shows various modifications of the RAV with different modulation depths at key points.

**Table 1.** Converter circuit modifications.

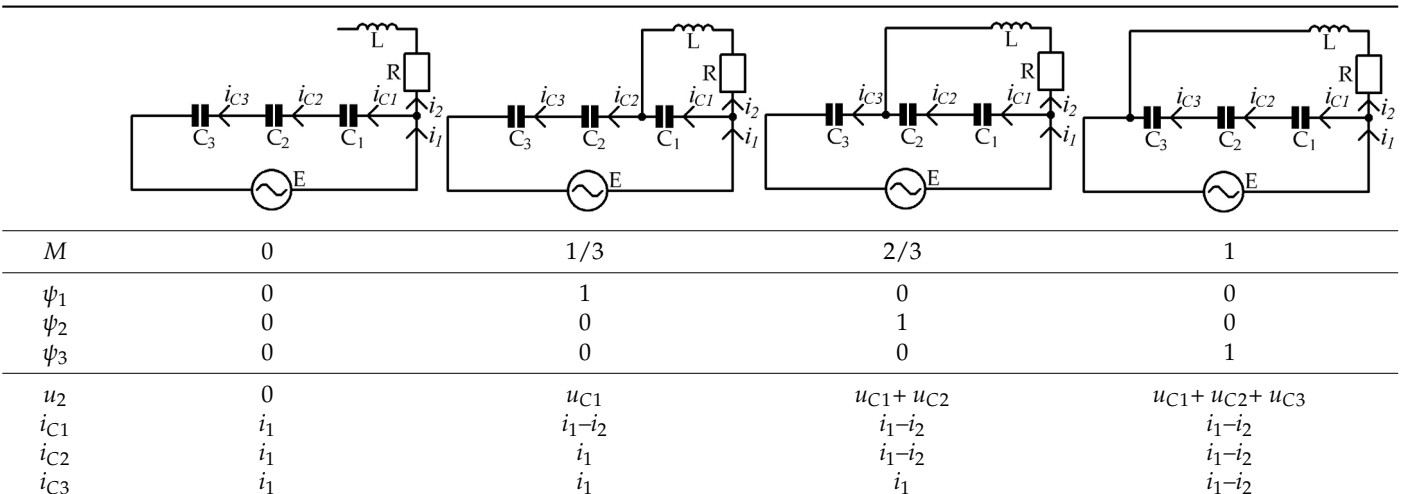

| M | 0 | 1/3 | 2/3 | 1 |
|---|---|---|---|---|
| $\psi_1$ | 0 | 1 | 0 | 0 |
| $\psi_2$ | 0 | 0 | 1 | 0 |
| $\psi_3$ | 0 | 0 | 0 | 1 |
| $u_2$ | 0 | $u_{C1}$ | $u_{C1} + u_{C2}$ | $u_{C1} + u_{C2} + u_{C3}$ |
| $i_{C1}$ | $i_1$ | $i_1 - i_2$ | $i_1 - i_2$ | $i_1 - i_2$ |
| $i_{C2}$ | $i_1$ | $i_1$ | $i_1 - i_2$ | $i_1 - i_2$ |
| $i_{C3}$ | $i_1$ | $i_1$ | $i_1$ | $i_1 - i_2$ |

Adding auxiliary switching functions $\psi$ based on the method of state variables will positively affect the optimization of the circuit's mathematical calculation and reduce the number of equations:

$$\begin{cases} u_2 = \psi_1 \cdot u_{C1} + \psi_2 \cdot (u_{C1} + u_{C2}) + \psi_3 \cdot (u_{C1} + u_{C2} + u_{C3}) \\ i_{C1} = i_1 - i_2 \cdot (\psi_1 + \psi_2 + \psi_3) \\ i_{C2} = i_1 - i_2 \cdot (\psi_2 + \psi_3) \\ i_{C3} = i_1 - i_2 \cdot (\psi_3) \end{cases} \tag{1}$$

The load voltage can be represented as the sum of the voltage across the resistor and the voltage across the inductor according to Kirchhoff's second law, and the voltage can be written in terms of currents (linear dependence for a resistor and differential dependence for an inductor):

$$u_2 = u_R + u_L = i_2 \cdot R + L \cdot \frac{d\,i_2}{dt} = i_2 \cdot R + L \cdot \overset{\bullet}{i_2} \tag{2}$$

Substituting the Equation (2) into the system (1) and describing the capacitance currents through capacitance voltage and the inductance voltage through inductance current through differential equations, we regroup the terms:

$$\begin{cases} L \cdot \overset{\bullet}{i_2} = u_{C1} \cdot (\psi_1 + \psi_2 + \psi_3) + u_{C2} \cdot (\psi_2 + \psi_3) + u_{C3} \cdot (\psi_3) + i_2 \cdot (-R) \\ C_1 \cdot \overset{\bullet}{u_{C1}} = i_1 + i_2 \cdot (-\psi_1 - \psi_2 - \psi_3) \\ C_2 \cdot \overset{\bullet}{u_{C2}} = i_1 + i_2 \cdot (-\psi_2 - \psi_3) \\ C_3 \cdot \overset{\bullet}{u_{C3}} = i_1 + i_2 \cdot (-\psi_3) \end{cases} \tag{3}$$

The idea is to introduce new variables in order to convert the second order equation into an equation of order one in higher dimensions. Using the method of describing the behavior of a dynamic system in the form of transfer functions by differential equations, we obtain the classical state space system:

$$\overset{\bullet}{x} = A \cdot x + B \cdot u \tag{4}$$

To simplify the calculation procedure, it is advisable to choose capacitors with the same capacitance (C). The capacitance calculation is based on two aspects: limiting the starting

current of the capacitors and assuming the switching frequency for a full charge to the supply voltage. We introduce auxiliary coefficients that facilitate further calculations ($\alpha$, $\beta$, $\gamma$):

$$\begin{cases} C_1 = C_2 = C_3 = C \\ \alpha = \dfrac{1}{L} \\ \beta = \dfrac{1}{C} \\ \gamma = \dfrac{R}{L} \end{cases} \tag{5}$$

We rewrite the system (3), taking into account the system (5):

$$\begin{cases} \dot{i}_2 = u_{C1} \cdot \alpha \cdot (\psi_1 + \psi_2 + \psi_3) + u_{C2} \cdot \alpha \cdot (\psi_2 + \psi_3) + u_{C3} \cdot \alpha \cdot (\psi_3) + i_2 \cdot (-\gamma) \\ \dot{u}_{C1} = i_1 \cdot \beta + i_2 \cdot \beta \cdot (-\psi_1 - \psi_2 - \psi_3) \\ \dot{u}_{C2} = i_1 \cdot \beta + i_2 \cdot \beta \cdot (-\psi_2 - \psi_3) \\ \dot{u}_{C3} = i_1 \cdot \beta + i_2 \cdot \beta \cdot (-\psi_3) \end{cases} \tag{6}$$

We rewrite the system (6) in equivalent matrix form:

$$\begin{bmatrix} \dot{i}_2 \\ \dot{u}_{C1} \\ \dot{u}_{C2} \\ \dot{u}_{C3} \end{bmatrix} = \begin{bmatrix} -\gamma & \psi_1 + \psi_2 + \psi_3 & \psi_2 + \psi_3 & \psi_3 \\ \beta \cdot (-\psi_1 - \psi_2 - \psi_3) & 0 & 0 & 0 \\ \beta \cdot (-\psi_2 - \psi_3) & 0 & 0 & 0 \\ \beta \cdot (-\psi_3) & 0 & 0 & 0 \end{bmatrix} \cdot \begin{bmatrix} i_2 \\ u_{C1} \\ u_{C2} \\ u_{C3} \end{bmatrix} + \begin{bmatrix} 0 \\ \beta \\ \beta \\ \beta \end{bmatrix} \cdot i_1 \tag{7}$$

In order to avoid differential equations, we decomposed the system of time dependence equations into active (index *a*) and reactive (index *r*) components in order to describe the matrix of unknowns and its derivatives, and also neglect the influence of phase components:

$$\begin{cases} i_{2(1)} = I_{2(1)a} \cdot \sin(w \cdot t) - I_{2(1)r} \cdot \cos(w \cdot t) \\ u_{C1(1)} = U_{C1(1)a} \cdot \sin(w \cdot t) - U_{C1(1)r} \cdot \cos(w \cdot t) \\ u_{C2(1)} = U_{C2(1)a} \cdot \sin(w \cdot t) - U_{C2(1)r} \cdot \cos(w \cdot t) \\ u_{C3(1)} = U_{C3(1)a} \cdot \sin(w \cdot t) - U_{C3(1)r} \cdot \cos(w \cdot t) \end{cases} \tag{8}$$

We took into account time derivatives:

$$\begin{cases} \dot{i}_{2(1)} = I_{2(1)a} \cdot w \cdot \cos(w \cdot t) + I_{2(1)r} \cdot w \cdot \sin(w \cdot t) \\ \dot{u}_{C1(1)} = U_{C1(1)a} \cdot w \cdot \cos(w \cdot t) + U_{C1(1)r} \cdot w \cdot \sin(w \cdot t) \\ \dot{u}_{C2(1)} = U_{C2(1)a} \cdot w \cdot \cos(w \cdot t) + U_{C2(1)r} \cdot w \cdot \sin(w \cdot t) \\ \dot{u}_{C3(1)} = U_{C3(1)a} \cdot w \cdot \cos(w \cdot t) + U_{C3(1)r} \cdot w \cdot \sin(w \cdot t) \end{cases} \tag{9}$$

Substituting the eighth and ninth systems into the seventh, we will see that to find the active and reactive values of the load current and voltages on capacitors, there is a system of four equations and eight unknowns (four active and four reactive components). To solve this system in order to compare the number of independent equations with the number of unknowns, as well as to get rid of time dependences in the form of harmonic functions, we multiply the previously obtained system of four equations in turn, first by a sinusoidal

time function and take the integral, taking into account averaging over the period, and then also by analogy with the cosine function:

$$
\begin{cases}
\frac{1}{T} \cdot \int\limits_{0}^{T} x_{(1)} \cdot \sin(w \cdot t)\, dt = \frac{X_{(1)a}}{2} \\
\frac{1}{T} \cdot \int\limits_{0}^{T} x_{(1)} \cdot \cos(w \cdot t)\, dt = -\frac{X_{(1)r}}{2} \\
\frac{1}{T} \cdot \int\limits_{0}^{T} \overset{\bullet}{x}_{(1)} \cdot \sin(w \cdot t)\, dt = \frac{X_{(1)r} \cdot w}{2} \\
\frac{1}{T} \cdot \int\limits_{0}^{T} \overset{\bullet}{x}_{(1)} \cdot \cos(w \cdot t)\, dt = \frac{X_{(1)a} \cdot w}{2}
\end{cases}
\tag{10}
$$

We take into account the sine component:

$$
\begin{bmatrix}
\frac{I_{2(1)r} \cdot w}{2} \\
\frac{U_{C1(1)r} \cdot w}{2} \\
\frac{U_{C2(1)r} \cdot w}{2} \\
\frac{U_{C3(1)r} \cdot w}{2}
\end{bmatrix}
=
\begin{bmatrix}
-\gamma & \psi_1 + \psi_2 + \psi_3 & \psi_2 + \psi_3 & \psi_3 \\
\beta \cdot (-\psi_1 - \psi_2 - \psi_3) & 0 & 0 & 0 \\
\beta \cdot (-\psi_2 - \psi_3) & 0 & 0 & 0 \\
\beta \cdot (-\psi_3) & 0 & 0 & 0
\end{bmatrix}
\cdot
\begin{bmatrix}
\frac{I_{2(1)a}}{2} \\
\frac{U_{C1(1)a}}{2} \\
\frac{U_{C2(1)a}}{2} \\
\frac{U_{C3(1)a}}{2}
\end{bmatrix}
+
\begin{bmatrix}
0 \\
\beta \\
\beta \\
\beta
\end{bmatrix}
\cdot i_1
\tag{11}
$$

Moreover, we take into account the cosine component:

$$
\begin{bmatrix}
\frac{I_{2(1)a} \cdot w}{2} \\
\frac{U_{C1(1)a} \cdot w}{2} \\
\frac{U_{C2(1)a} \cdot w}{2} \\
\frac{U_{C3(1)a} \cdot w}{2}
\end{bmatrix}
=
\begin{bmatrix}
-\gamma & \psi_1 + \psi_2 + \psi_3 & \psi_2 + \psi_3 & \psi_3 \\
\beta \cdot (-\psi_1 - \psi_2 - \psi_3) & 0 & 0 & 0 \\
\beta \cdot (-\psi_2 - \psi_3) & 0 & 0 & 0 \\
\beta \cdot (-\psi_3) & 0 & 0 & 0
\end{bmatrix}
\cdot
\begin{bmatrix}
-\frac{I_{2(1)r}}{2} \\
-\frac{U_{C1(1)r}}{2} \\
-\frac{U_{C2(1)r}}{2} \\
-\frac{U_{C3(1)r}}{2}
\end{bmatrix}
+
\begin{bmatrix}
0 \\
\beta \\
\beta \\
\beta
\end{bmatrix}
\cdot i_1
\tag{12}
$$

Combining system (11) with system (12), we can obtain the final matrix system with eight unknowns. To find the resulting amplitude of any variable and any harmonic, it is necessary to add the vectors of the active and reactive components on the complex plane:

$$
\overrightarrow{X_{i(1)}} = \overrightarrow{X_{i(1)a}} + \overrightarrow{X_{i(1)r}}
\tag{13}
$$

Moreover, taking into account the orthogonality of the components, we obtain a universal formula for calculating the effective values of the first harmonic of all unknown elements:

$$
X_{i(1)} = \sqrt{X_{i(1)a}{}^2 + X_{i(1)r}{}^2}
\tag{14}
$$

The differential equations algebraization method underlies the calculation of the effective values of the first harmonics of currents and voltages of the circuit under study and hence the analysis of electromagnetic processes. The accuracy of the solutions obtained on the basis of the DEA method varies depending on the degree of accuracy in redefining these same algebraic equations and is accordingly proportional to the level of the assumption. The advantage of using this method lies in avoiding the calculation of all trigonometric expressions.

High frequency (HF) PWM and sawtooth reference voltage generator with rising slope is used to control the converter. For three-zone regulation, the signals of the reference signals are mathematically described by the following formulas:

$$
\begin{cases}
s_{\text{ref 1}}(t) = \frac{1}{6} + \frac{1}{3 \cdot \pi} \cdot \text{arctg}(\text{tg}(\pi \cdot f_{\text{sw}} \cdot t)) \\
s_{\text{ref 2}}(t) = \frac{3}{6} + \frac{1}{3 \cdot \pi} \cdot \text{arctg}(\text{tg}(\pi \cdot f_{\text{sw}} \cdot t)) \\
s_{\text{ref 3}}(t) = \frac{5}{6} + \frac{1}{3 \cdot \pi} \cdot \text{arctg}(\text{tg}(\pi \cdot f_{\text{sw}} \cdot t))
\end{cases}
\tag{15}
$$

For a visualization of the control system operation principles, work on the formation of control impulses in dynamics will be considered, i.e., with a linear increase in the modulating signal (modulation depth *M*) in Figure 6. The duration of the opening of each zone corresponds to one period of the main voltage (20 ms).

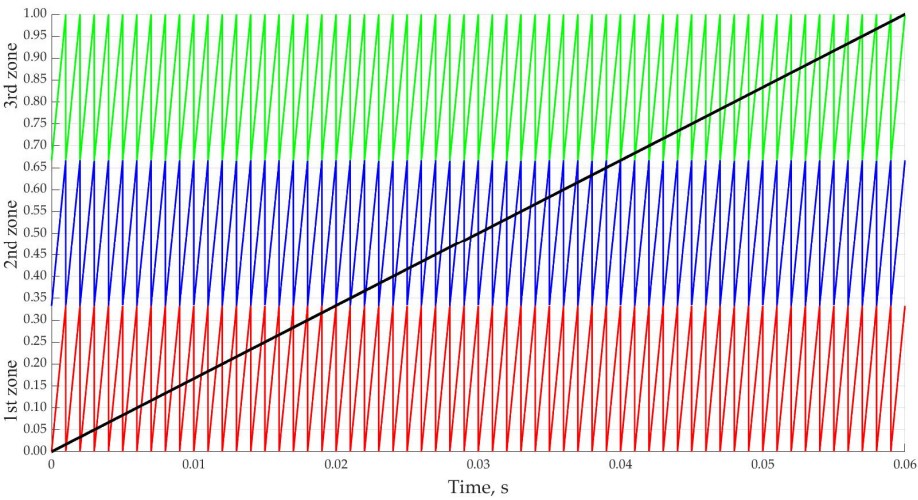

**Figure 6.** Oscillograms of modulating and reference signals in dynamics.

The principle of PWM operation is based on comparing two signals (modulating and reference) and generating a control signal (pulse) for power switches. The formation of control pulses in Figure 7, both in statics and dynamics, occurs after applying three pairs of signals to comparators (operational amplifiers (OA) without feedback). Already at the stage of experimental stand development, it can be taken into account that the power supply of the OA must correspond to a level of sufficient voltage for both locking the power switch and unlocking it.

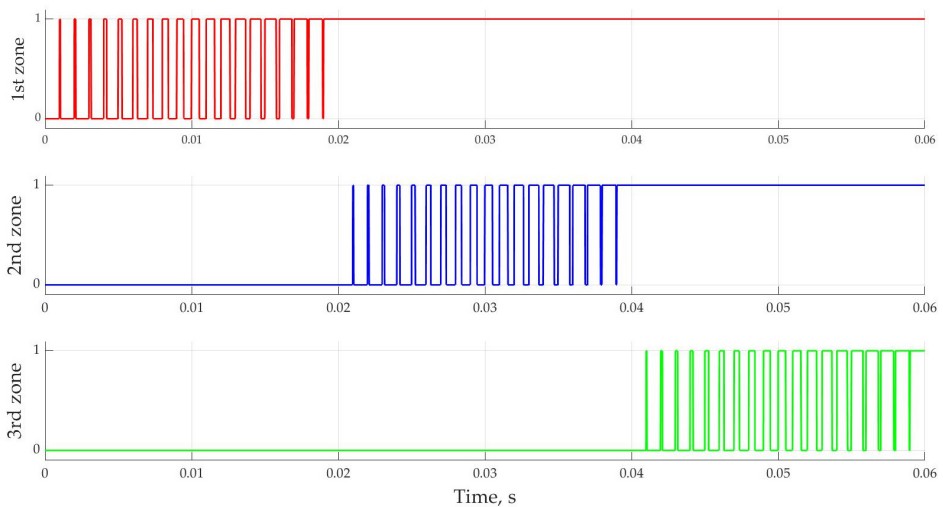

**Figure 7.** Oscillograms of control pulses in dynamics.

Switching of power switches is implemented according to a special algorithm to prevent short-circuiting of capacitive elements when zones are opened one by one. This principle will have a positive effect on the implementation of the MPCS since the control characteristic will be monotonically increasing over the entire range of modulation depth. However, this approach requires the presence of a network synchronization unit (NSU) when generating control pulses, which means that a voltage sensor is used. It is this reason that determines such a choice of semiconductor switches of bidirectional conductivity with a connected diode.

Sequential opening of zones by increasing the depth of modulation contributes to a change in shape in Figure 8 and an increase in the effective value of the load voltage. After the opening of the first zone (M > 1/3), the voltage across the capacitor $C_1$ is applied to the load (in red). After opening the second zone (M > 2/3), the voltage across the capacitor $C_2$ is applied to the load (in blue). After opening the third zone (M = 1), the voltage on the capacitor $C_3$ is also supplied to the load (in green). It is possible to control the converter not according to the principle of the sequential opening of zones. In this case, using the example of opening the second zone, it is possible to obtain not from 1/3 to 2/3 of the mains voltage but from 0 to 2/3.

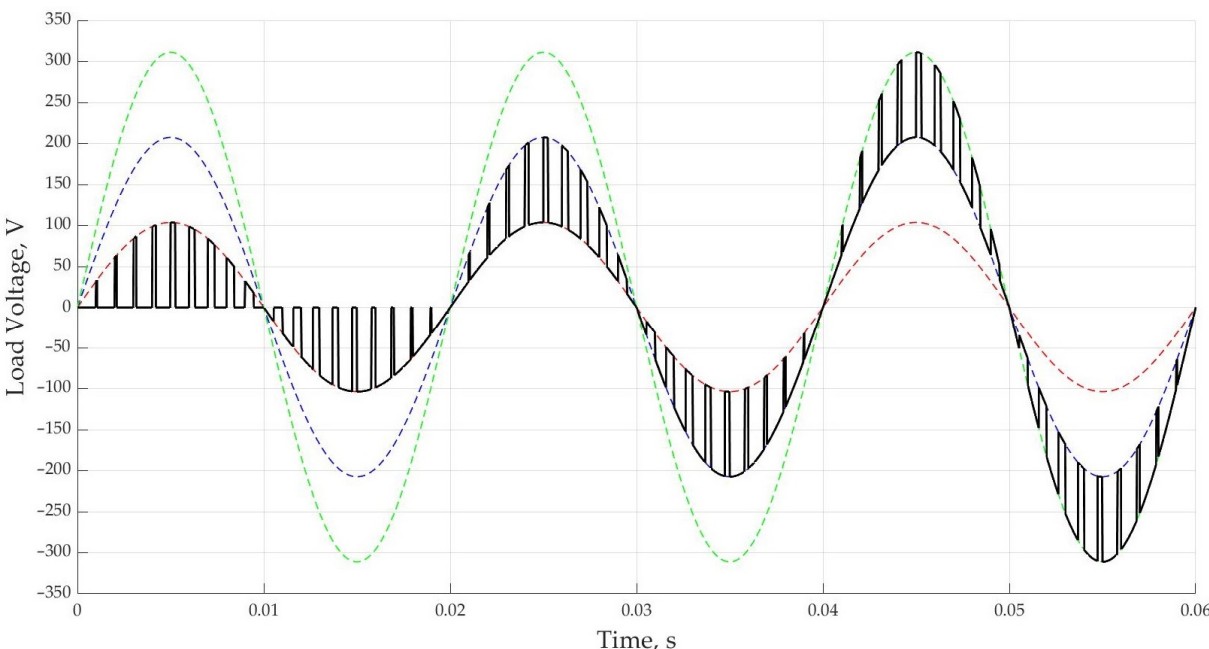

**Figure 8.** Oscillograms of load voltage in dynamics.

The fundamental harmonic of the load voltage should make the main contribution to the spectral composition when using the FT. With an increase in the switching frequency, it is possible to achieve an increase in its multiplicity with the frequency of the supply voltage, which means moving the minor switching harmonics away from the initial main one, thereby reducing their impact on quality. According to the FT, when decomposing the signal under study into harmonic components with increasing frequency, there should be a tendency to decrease the amplitudes of higher harmonics according to the Dirichlet principle.

At a switching frequency of 1 kHz, the multiplicity with a supply voltage frequency of 50 Hz will be equal to 20. Accordingly, the influence of non-fundamental harmonics will be greatest at odd frequencies near multiples of 20 (19 and 21). Moreover, switching harmonics are characteristic only for odd multiplicity (1–20, 2–60, 3–100, etc.), as shown in Figure 9.

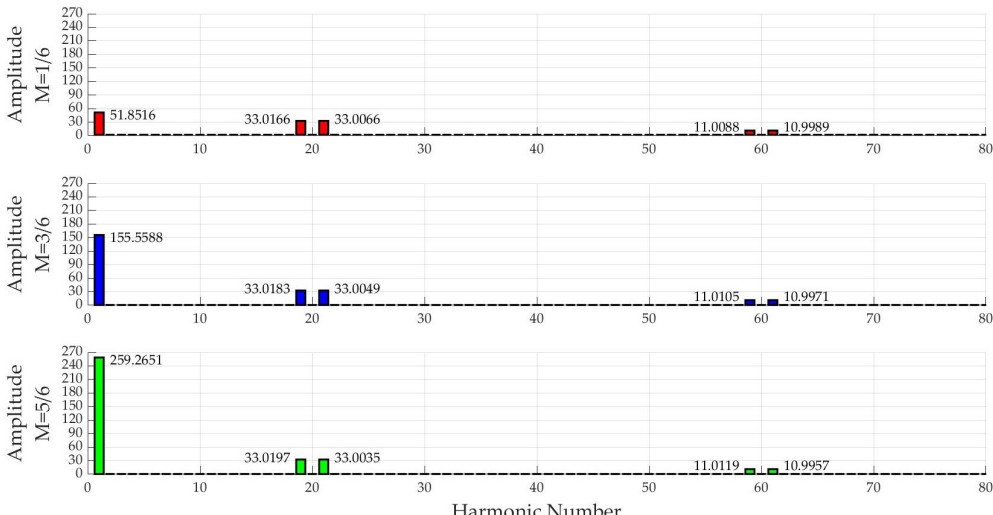

**Figure 9.** Spectral composition of load voltage at different modulation depths.

## 4. Simulation

Simulation modeling of a multi-zone AC voltage converter as a soft starter for an induction motor was carried out in Matlab/Simulink. The load is a single-phase induction motor. Simulation parameters are presented in Table 2.

**Table 2.** Simulation Parameters.

| Name | Designation | Value |
|---|---|---|
| IM power, kW | $P_{load}$ | 1.6 |
| Mains voltage, V | $V_g$ | 220 |
| Network frequency, Hz | $f_g$ | 50 |
| Switching frequency, Hz | $f_{sw}$ | 1000 |
| Number of zones | $N$ | 3 |
| Capacitance of the 1st capacitor, μF | $C_1$ | 200 |
| Capacitance of the 2nd capacitor, μF | $C_2$ | 200 |
| Capacitance of the 3rd capacitor, μF | $C_3$ | 200 |
| Nominal IM current, A | $I_m$ | 10 |
| Nominal IM speed, rpm | $w$ | 1500 |
| PF | $cos(\varphi)$ | 0.84 |

When the motor is connected directly to the mains, large starting currents occur, up to five times the rated value (500%) in Figure 10. However, the acceleration time is 0.15 s.

The harm from a fivefold excess of the rated value motor starting current was described earlier. In order to solve this problem, it was decided to use the principle of a smooth increase in power when starting the motor. A dynamic increase in the modulation depth should, by means of regulation by a multi-zone AC voltage converter, reduce the magnitude of the starting currents. To ensure that the starting current is limited, the motor current feedback in the relay controller is used. The principle of operation is based on the disconnection of all zones (M = 0) in case of exceeding the rated effective value of the motor current in Figure 11. The principle of operation of such a relay controller is based on logical operations and is quite simple in software implementation in practice. The acceleration duration was set to 4 s, and the motor reached its nominal speed already 3 s after the start, but the main significant acceleration began after 1 s. The blue color indicates the root mean square (RMS) value of the motor current.

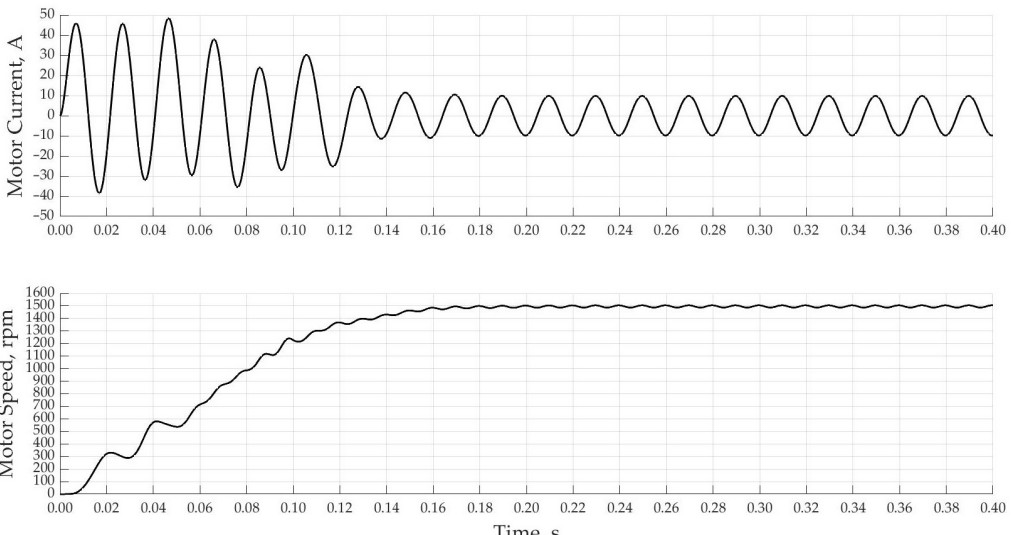

**Figure 10.** Starting current and motor speed under direct start.

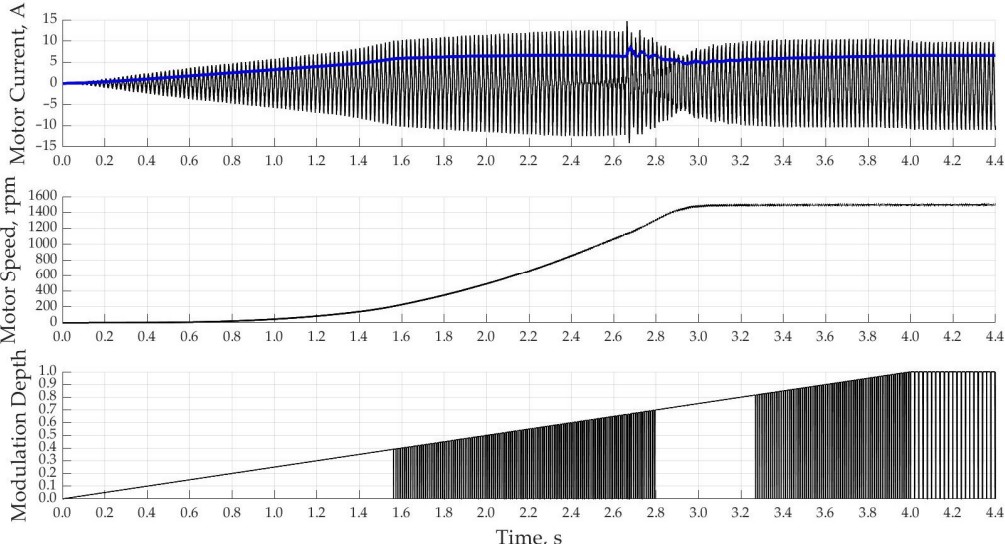

**Figure 11.** Starting current, motor speed and modulation depth under relay controller with all zones disabled.

A significant disadvantage of using such a CS is the poor quality of the motor current in Figure A1. The use of a relay control controller with disconnection of not all zones (M = 0) but with disconnection of only the last zone (M = M − 1/3) only in case of exceeding the starting current of the motor is shown in Figure 12.

This method slightly improves the shape of the consumed current by smoothing out the switching ripple but is still unsatisfactory in Figure A2. It would be advisable to abandon the idea of using a relay regulator and change the controller structure to a proportional-integral one.

When using a PI-controller with a single-loop system of automatic control of the motor current with a rated current according to the setting, the acceleration of the motor starts much faster and reaches the rated speed already in 2.5 s in Figure 13. This control method is characterized by overcurrent at speeds close to the nominal ones due to the fact that the regulator itself has a delay time to regulate the magnitude of the output voltage connected to the motor.

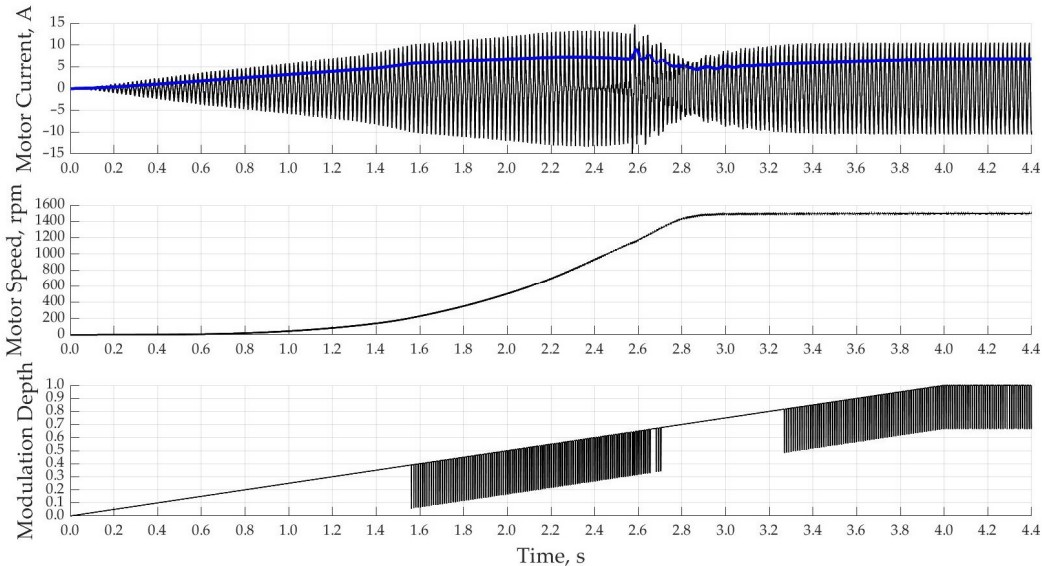

**Figure 12.** Starting current, motor speed and modulation depth under relay controller with the last zone disabled.

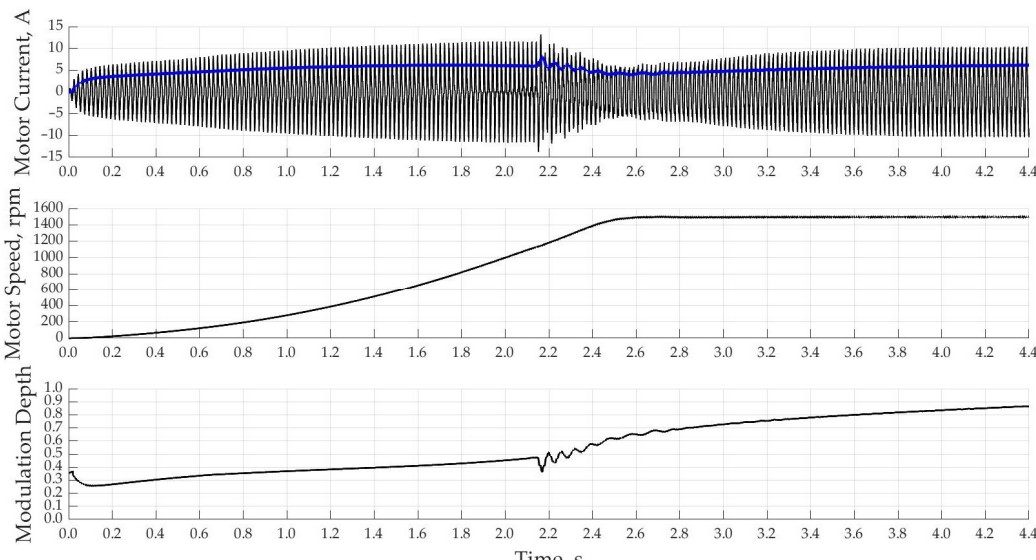

**Figure 13.** Starting current, motor speed and modulation depth under the PI-controller with single-loop CS of IM current with rated current according to reference.

When using a PI-controller with a single-loop automatic motor current CS with a linear increase in current by setting, the acceleration of the motor starts much longer and reaches the rated speed in 3.6 s in Figure 14. The shape of the motor speed is not optimal. This control method is characterized by a very slow increase in current and, as a result, a long-time idle IM.

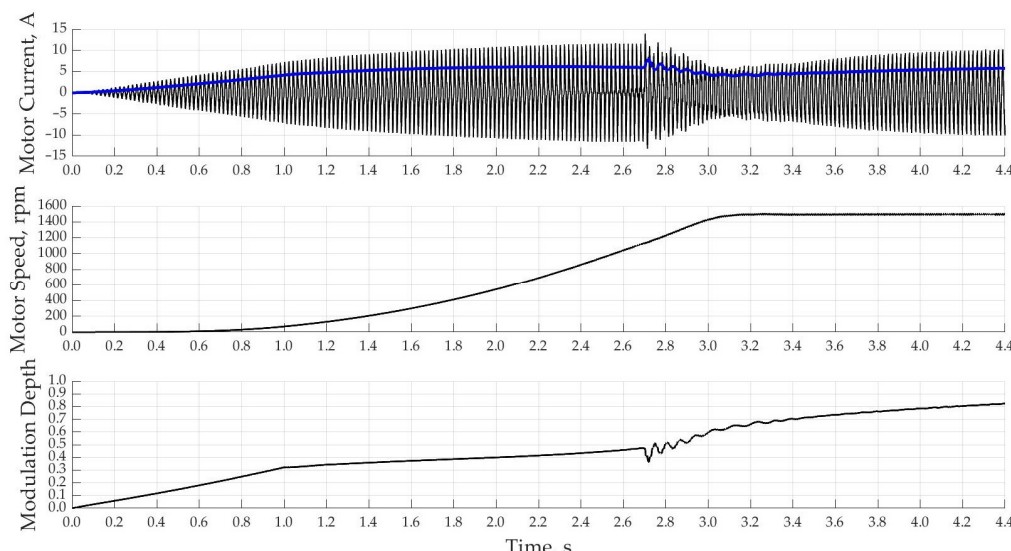

**Figure 14.** Starting current, motor speed and modulation depth with PI-controller with single-loop CS of IM current with linear increase in current by reference.

When using PI controllers with a double-loop system for automatic control of the current and motor speed by setting, it should ensure the acceleration of the motor according to the desired form. Two types of settings for the desired RPM will be considered. First is linear acceleration; its main feature is the ease of implementation of data for motor acceleration; it will be enough just to specify the time. The correct operation of the converter as a soft starter is determined by its adaptive structure and the introduction of a double-circuit automatic CS for starting current and motor speed in Figure 15.

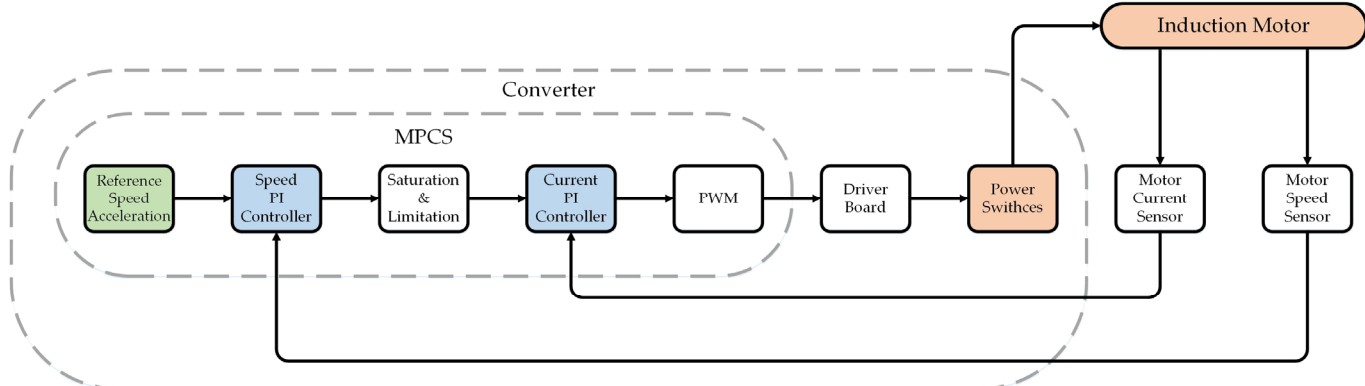

**Figure 15.** Block diagram of connecting converter to induction motor.

Since the output signal of the external and slower circuit for setting the desired motor speed is the input of the internal and faster circuit for setting the starting current, it is advisable to use a limiter to the nominal RMS value. Figure 16 shows the results of simulation modeling with the proposed control algorithm.

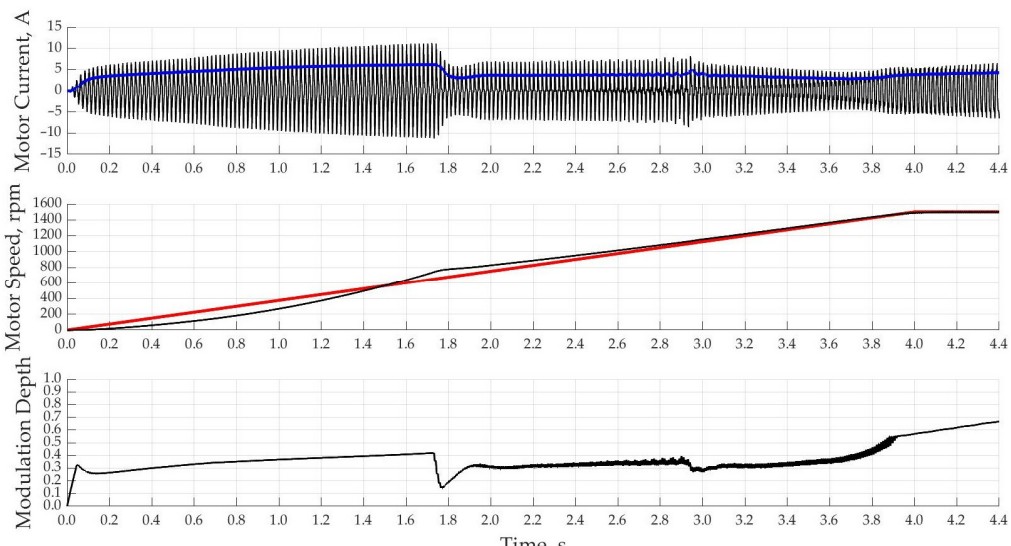

**Figure 16.** Starting current, motor speed and modulation depth for case of PI-controller with double-loop CS of current and speed of IM with linear increase in speed by reference.

It can be seen from the simulation results that up to a time of 1.4 s, the real induction motor speed remained from the external loop setting signal, which means that the value of the error along the speed loop has a negative sign. This is due to too fast acceleration at low motor speeds. After a time of 1.4 s, the real IM speed is already ahead of the reference signal along the outer loop, which means that the value of the error along the speed loop has a positive sign. As a result, the converter control signal reduces the modulation depth, thereby reducing the induction motor supply voltage. The transient process is characterized by an overshoot.

In order to ensure that the value of the external loop error in terms of motor speed is close to zero, it is advisable to change the form of acceleration from linear to sigmoidal. The choice of just such a form of reference IM speed is characterized by the largest value of the derivative in the middle of acceleration. In simple words, there will be a smoother speed setting in the first half of acceleration, and in the second half of acceleration, a sharper one. The results of such a CS are presented in Figure 17.

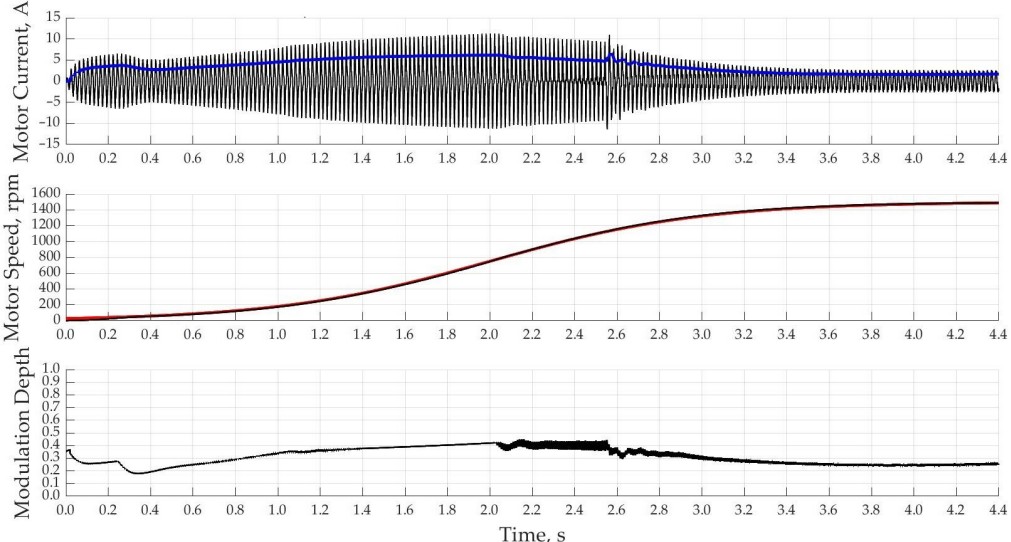

**Figure 17.** Starting current, motor speed and modulation depth for case of PI-controller with double-loop CS of current and speed of IM with sigmoidal increase in speed by reference.

The simulation of the proposed double-loop CS for starting current and motor speed with a sigmoidal setting provides the most optimal acceleration in 4 s. However, when the motor shaft is loaded, there is a risk of overturning, which will be clearly shown in Figure 18 by a negative speed value of up to 1.2 s.

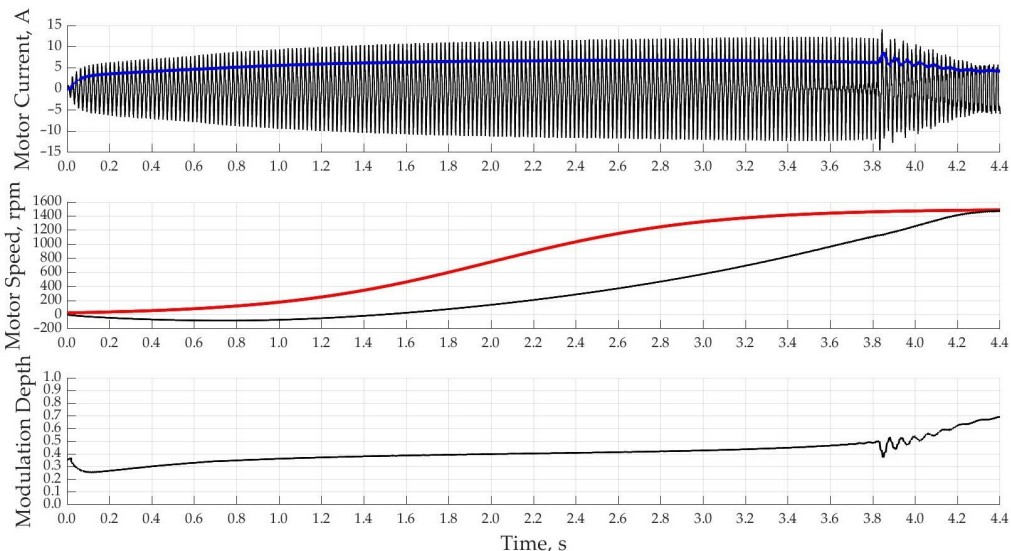

**Figure 18.** Starting current, motor speed and modulation depth for case of PI-controller with double-loop CS of IM current and speed with sigmoidal increase in speed by reference and loaded shaft.

If this algorithm is used to accelerate the IM with a load on the shaft that will linearly increase to the nominal value in 2 s, then optimal acceleration without overturning is expected. However, in the case of reaching the rated torque of the motor load in 2 s, there is a lag between the actual IM speed and the desired ones in Figure 19. Motor acceleration with a smooth increase in torque is typical for fan systems.

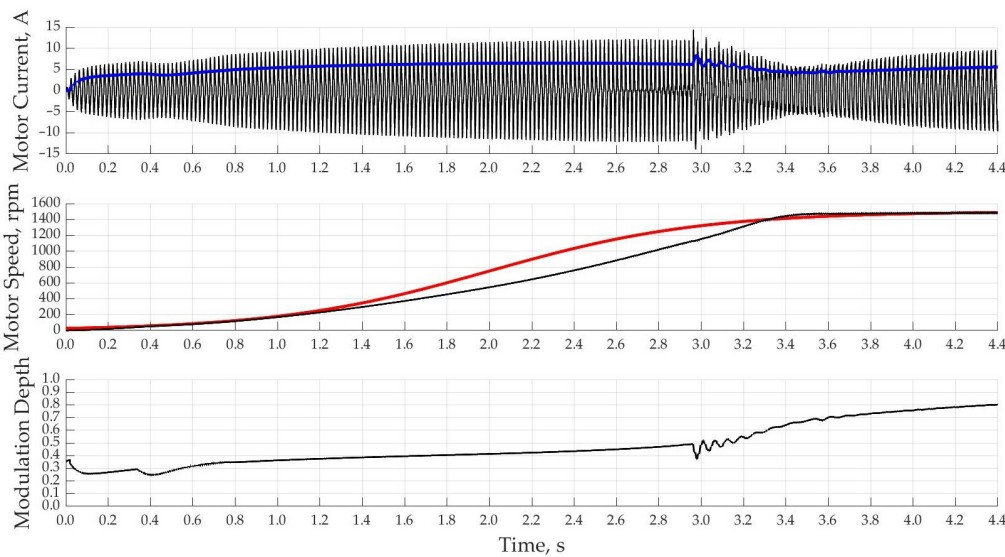

**Figure 19.** Starting current, motor speed and modulation depth for case of PI-controller with double-loop CS of IM current and speed with sigmoidal increase in speed by reference and smoothly loaded shaft.

The use of a multi-zone AC voltage converter with a double-loop CS for starting current and motor speed as a soft starter for IMs shows that it is possible to both limit the starting current and set the desired shape and duration of the motor acceleration.

## 5. Experiment

To check the correct operation of the multi-zone AC voltage converter in practice, an experimental stand was implemented in Figure 20. At this stage of the study, not an induction motor but an active load was used as a load. The formation of control pulses is based on information about the sign of the mains voltage, which means that in order to implement the NSU, it is necessary to connect a step-down transformer, the output voltage of which is supplied to the OA board. A current sensor was used to obtain current load data, and a voltage sensor was used to obtain the mains voltage sign. The formation of the reference voltage generator was implemented in the MPCS. The mains voltage was connected to the converter through a control autotransformer.

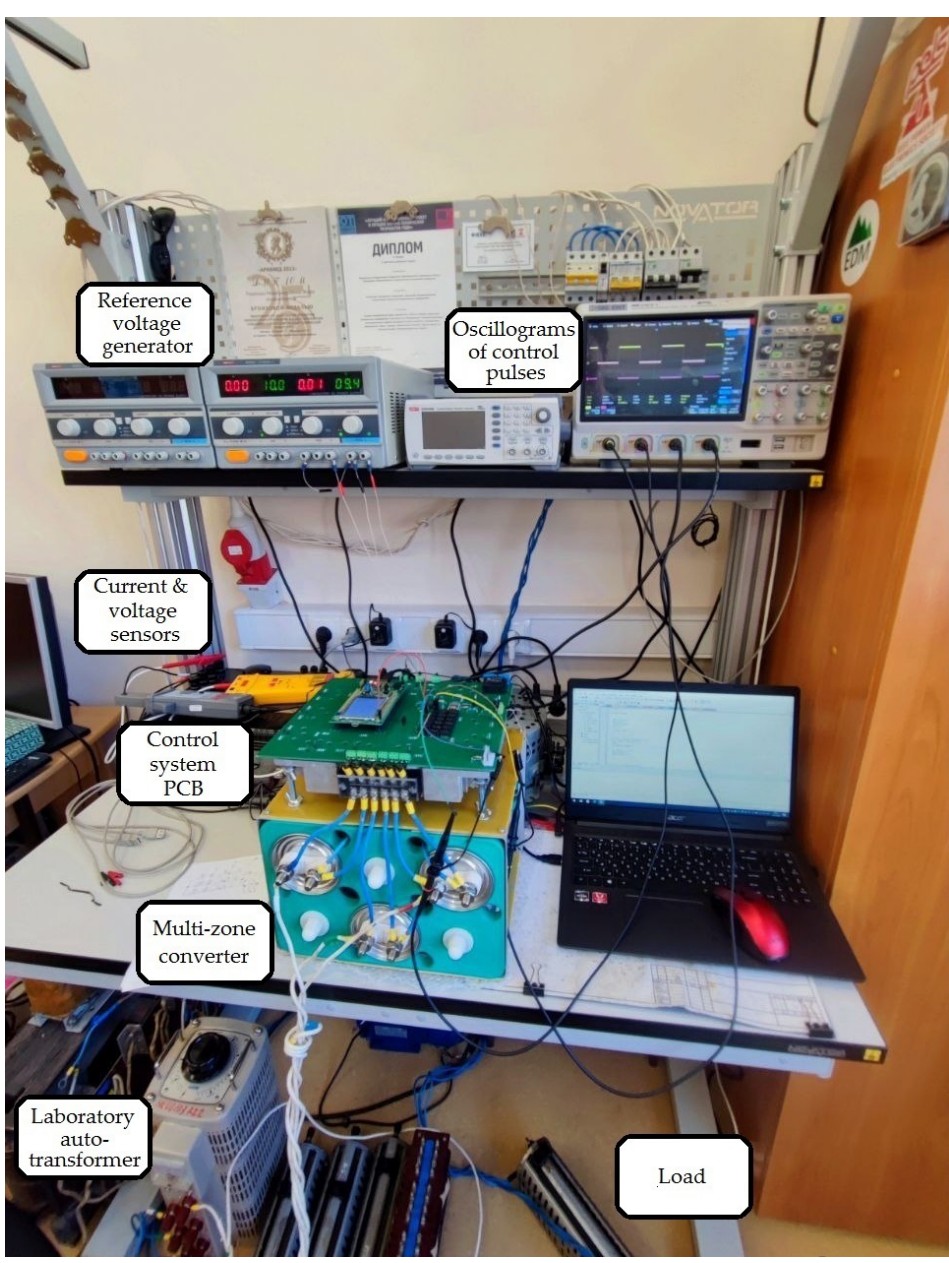

**Figure 20.** Photo of experimental stand.

The amplitude of the MPCS pulse meander at the output is 3.3 V. Control pulse formation when changing the modulation depth is shown in Figure 21. To use transistors as power switches, it is necessary to increase the amplitude of control pulses by means of the OA board.

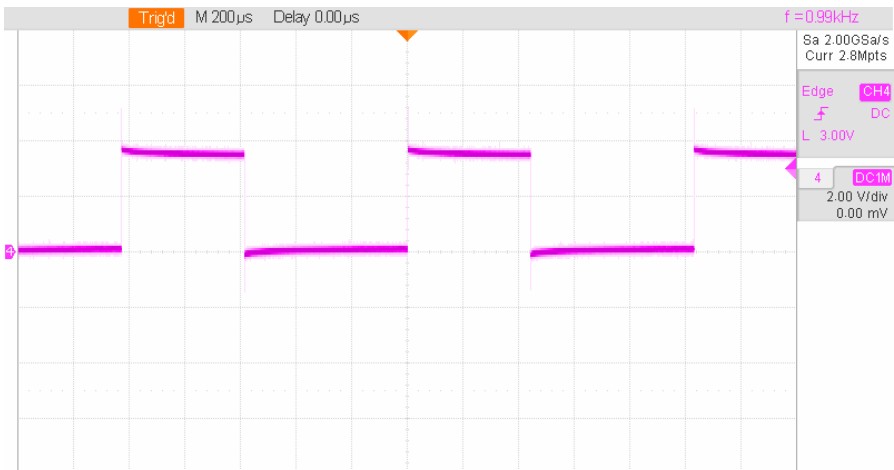

**Figure 21.** Oscillogram of control pulses.

The shape of the load voltage in the middle of the first zone is shown in Figure 22.

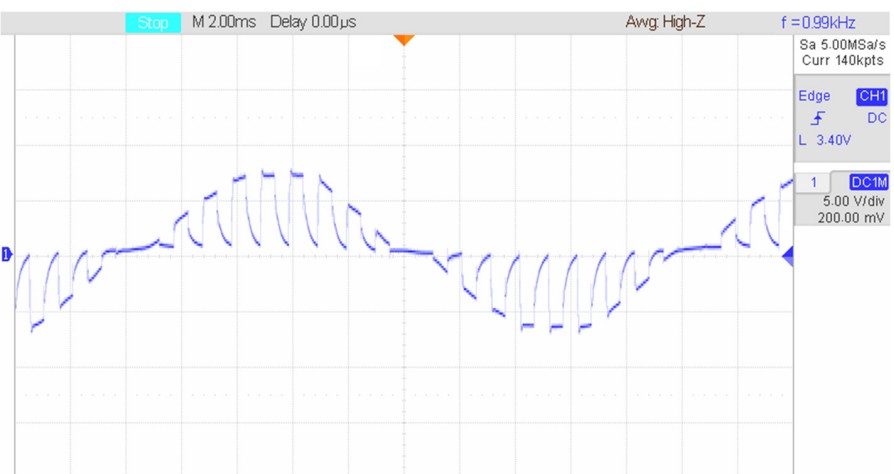

**Figure 22.** Output voltage waveform.

Figure 23 shows the current, leading relative to the mains voltage. It results from the capacitance in the converter structure. This, in turn, improves EMC.

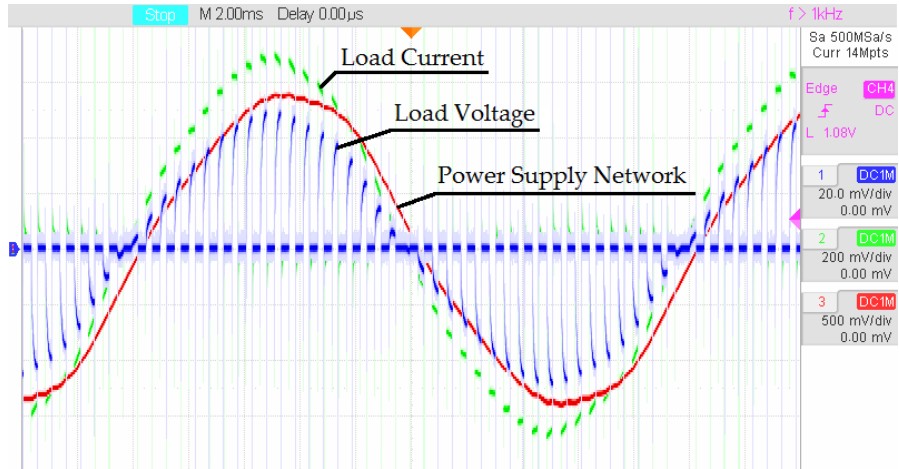

**Figure 23.** Oscillograms of load voltage and current under active load.

Switching harmonics appear in the voltage spectrum in Figure 24. The trend towards a decrease in switching harmonics with an increase in their number remains.

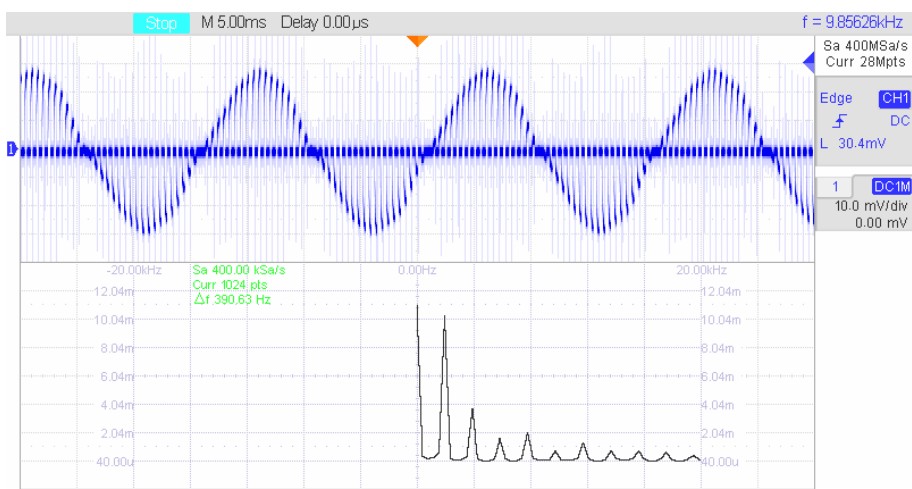

**Figure 24.** Load voltage spectrum.

## 6. Discussion and Conclusions

The proposed multi-zone AC voltage converter can be used as a soft starter for IM. The circuit design of a multi-zone AC voltage converter is possible both in a single-phase version and in a three-phase one. When regulating the RMS load voltage, the VCF lies in the range of 0 to 1. The principle of RAV operation is based on a smooth increase in power at low motor speeds. The presence of feedback on the starting current and motor speed provides the desired form of acceleration time dependence without exceeding the nominal current values. The use of transistors as power switches makes it possible to apply an HF PWM in an MPCS, which significantly reduces the effect of higher harmonics and improves the spectral composition of the output signal. The structure of the investigated converter is transformerless; thus, the type of switching is changed from mechanical to electric. Replacing the transformer voltage divider with a capacitive one will also reduce the weight and size of the converter. The use of a multi-zone regulator is relevant for high-voltage systems. Increasing the number of zones will improve the spectral composition of the output voltage.

An algorithm for a double-loop automatic CS for a multi-zone RAV has been developed, which makes it possible to minimize the influence of asynchronous motors on the mains voltage, and therefore improve EMC. The use of capacitive voltage dividers will also compensate for the consumed reactive power. The features of modern soft starters are analyzed. For the proposed converter, its structure and operation principle are described. A mathematical calculation was carried out by the DEA method. Simulation modeling was performed in the Matlab/Simulink environment. The adequacy and correctness of the RAV operation were tested on an experimental stand, which will also be used for further research when connecting IM as a converter load. Thus, the study of a multi-zone AC voltage converter contributes to the development of single-stage regulators in high-voltage systems.

## 7. Patents

1. Pat. 2373625, Variable Voltage Zoned Regulator, Zinoviev G.S. Russian Federation, IPC N 02 M 5/257, applicant and patentee NSTU—No. 2008138763/09; declared 29/09/08; publ. 20.11.09, Bul. No. 32. Status: valid.

2. Pat. 2368937, AC Voltage Controller, Zinoviev G.S. Russian Federation, IPC N 02 M 1/20, applicant and patentee NSTU—No. 2008130246/09; declared 21/07/08; publ. 27.09.19, Bul. No. 27. Status: valid.

**Author Contributions:** Conceptualization, E.K.; methodology, E.K., A.U. and G.Z.; software, E.K.; validation, E.K., A.U., G.Z. and N.L.; formal analysis, E.K.; investigation, E.K. and A.U.; resources, E.K. and A.U.; data curation, E.K. and A.U.; writing—original draft preparation, E.K.; writing—review and editing, E.K.; visualization, E.K.; supervision, A.U., E.G. and R.S.; funding acquisition, E.K. and A.U. All authors have read and agreed to the published version of the manuscript.

**Funding:** This research was funded by NSTU development program, scientific project No. C22-23.

**Acknowledgments:** The authors are grateful to the Department of Electronics and Electrical Engineering and the Institute of Power Electronics for their contributions and the supporting this research, as well as to Novosibirsk State Technical University for financial support.

**Conflicts of Interest:** The authors declare no conflict of interest.

## Abbreviations

| | |
|---|---|
| ANN | Artificial Neural Network |
| DEA | Differential Equations Algebraization |
| CS | Control System |
| EMC | Electromagnetic Compatibility |
| FT | Fourier Transform |
| HF | High Frequency |
| IM | Induction Motor |
| LF | Low Frequency |
| MLVSI | Multi-level Voltage Source Inverter |
| MPCS | Microprocessor Control System |
| NSU | Network Synchronization Unit |
| OA | Operational Amplifier |
| PF | Power Factor |
| PWM | Pulse Width Modulation |
| RAV | AC Voltage Regulator |
| RMS | Root Mean Square |
| THD | Total Harmonic Distortion |
| VCF | Voltage Conversion Factor |

## Appendix A

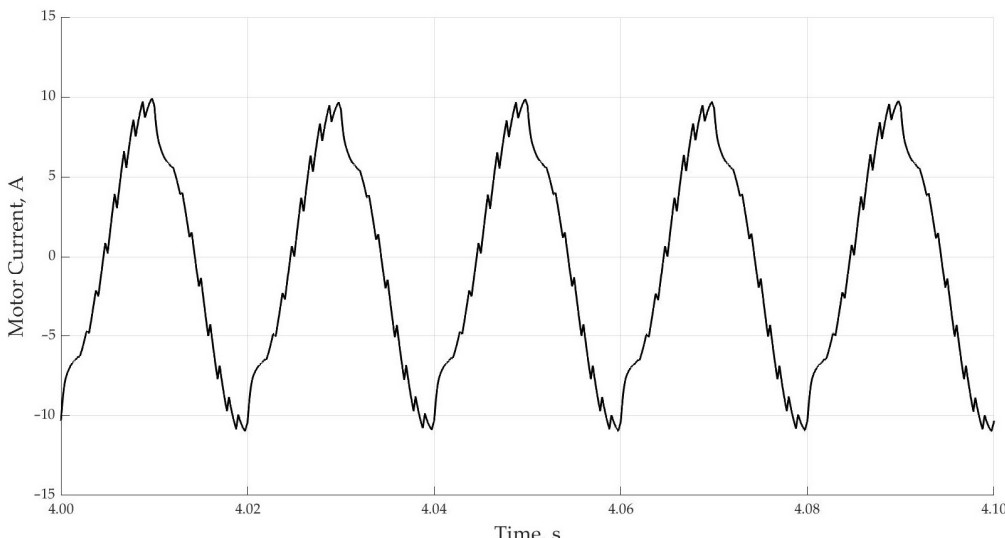

**Figure A1.** Motor current waveform for case of using relay controller with all zones disabled.

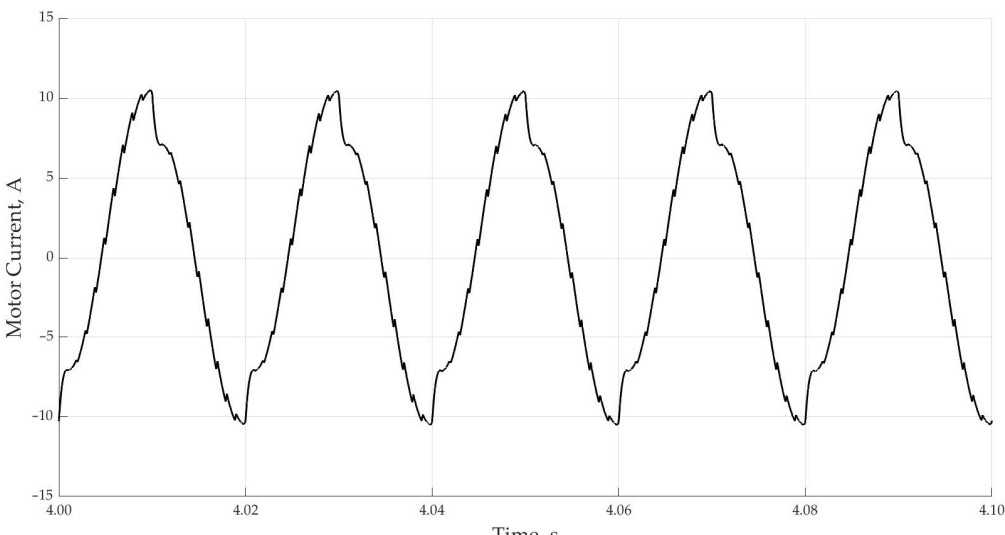

**Figure A2.** Motor current waveform for case of using relay controller with last zone disabled.

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
