# Peer review of "Analysis of the Control System for a Soft Starter of an Induction Motor Based on a Multi-Zone AC Voltage Converter"

_electronics, doi:10.3390/electronics12010056_

Round 1

Reviewer 1 Report

1) Line 41: "Low quality is understood..." It would be better written as 'Low power quality.'

2) Line 190-194: The structures on these sentences need to be edited for better clarity. 

3) Table 2: The heading for the third column should be changed from 'meaning' to 'values.'

4) Figures 23 and 24: It would be meaningful if the experimental results for the load voltage and current can be compared with the simulation in terms of the THD and power factor.  

5) In line 451, in what way the capacitors improved the EMC? Is there any measurements done on this? 

6) The conclusion can be improved by adding a few sentences on the impact of this work to this research area.

Author Response

Good afternoon!

We express our gratitude to you for the review of our article.

Below are responses to comments.

1) We added "low voltage quality".

2) We made changes to the sentences and their order in the text, and also rewrote their structures.

3) We changed "meanings" to "values".

4) We fully agree with your proposal and share the proposed logic of the study, however, our measuring instruments only provide an opportunity to display the spectral composition without displaying signal quality indicators. This is a very valuable note and we will adhere to it in future developments.

5) We have restructured the sentence with the description of Figure 23 to describe the principle of EMC improvement. Quality indicators of EMC improvement are not presented, only an indirect analysis was carried out based on the shift of the oscillograms with the connection of the converter.

6) We improved the inference by adding a scope impact clause.

In addition, together with the translators, we have also changed the wording of the text for greater clarity, thereby improving the level of English.

Thanks for your expert review!

Reviewer 2 Report

This manuscript has some drawbacks in the organization aspect. Its current form is not suitable for publication.

(1)   The introduction is not in a comprehensive way. The authors should focus more on the control technologies. What’s the new challenges in the control technology as the RAV technology updated?

(2)   In the section 3 “Mathematical Calculation”, many equations are common sense.

(3)   The authors presented many waveforms in the manuscript. But direct performance indicators are also necessary for the analysis of their performances.

(4)   How does the figures in experiment part correspond to the figures in simulation part?

(5)   Are there any negative influences when the zone number increases?

Author Response

Good afternoon!

We express our gratitude to you for the review of our article.

Below are responses to comments.

1) We have made a change in the structure of the introduction and annotations. We have added sentences that reflect the problem statement and the shortcomings of existing analogues. The purpose of this work was to study the control algorithms for a soft starter for induction motors based on a multi-zone AC voltage converter. The basic principle of operation of the regulator is based on a smooth increase in power, and not the frequency of the input voltage. The disadvantage is the fact that in production the most priority control method is the use of frequency converters, which in turn requires the presence of a rectifier and an inverter in the structure of the device.

2) We tried to describe mathematical calculations in detail, since the considered differential equations algebraization (DEA) method is rarely found in the scientific literature of this field of study.

3) We agree that much attention is paid to the oscillograms of soft start and acceleration of an induction motor with various control algorithms for a multi-zone AC converter. The starting current limitation acts as indicators of quality and efficiency of absolutely all control methods, and for systems with a two-loop control system, there is also a zero error in the external loop of the PI controller in terms of motor speed. These 2 indicators are fundamental in the development of the proposed soft starter.

4) In the experimental part, a control algorithm for a multi-zone converter is implemented when connected to an active load. The purpose of this article was not to compare the results of simulation modeling in dynamics with mathematical calculation and experiment in static mode. The experiment contains oscillograms of the transducer operation in the middle of the first zone in statics (link to figure 8 when opening the first zone in dynamics) and spectral composition (link to figure 9).

5) An increase in the number of converter zones increases not only the number of reactive elements (capacitors), but also the number of power switches, which in turn increases the dynamic losses during switching and reduces efficiency. That is why the topology of the regulator with 3 zones was considered, because its control characteristic has a slight change as well as the THD dependence in comparison with 4 zones. By the way large numbers of zones are useful and actual for high voltage systems, that didn’t considered at this article.

In addition, together with the translators, we have also changed the wording of the text for greater clarity, thereby improving the level of English.

Thanks for your expert review!

Reviewer 3 Report

Abstract:

- problem statement is not strong. improve it. 

- Summary of findings is also not really found in the abstract. 

Intro:

- objectives of this paper not really mention.

- details review of past research or methods including performance characteristics such as power losses, response time, accuracy etc. 

section 2:

- there is no details explanation of previous techniques as references for the proposed technique. the author should add also previous techniques so readers can differentiate between the existing and proposed techniques directly.

Section 3:

- ok

section 4:

- Sampling time of simulation missing and also switching frequencies

- what is the accuracy and power losses of this proposed technique? 

- It is good to put all the results in one table. 

Section 5:

- what is the accuracy and power losses of this proposed technique. 

- It is good to put all the results in one table. 

Conclusions:

ok 

Author Response

Good afternoon!

We express our gratitude to you for the review of our article.

Below are responses to comments.

Annotation:

1) We improved the formulation of the problem and added a description of the shortcomings in the methods used in practice to deal with this problem (frequency converters).

2) A summary of the conclusions is presented in the first sentence of the conclusion. We consider it obvious that the problem posed can be solved with the help of our development, and the multi-zone AC voltage converter can be used as a soft starter for induction motors.

Introduction:

1) The objectives of the article are presented in the penultimate paragraph of the introduction section.

2) We did not conduct a detailed literature review comparing quantitative quality indicators. This was not the aim of the study.

Section 2:

1) The paper presents the most detailed analysis of the mathematical calculation, each action is explained and commented. In addition, references to the proposed short calculation method (DEA) are presented in our previous works [39–42]. The purpose of the article was not to compare different methods of mathematical calculation. We will take into account your remark in our further works devoted specifically to the comparison of calculation methods.

Section 4 and 5:

1) Information about the simulation time can be obtained from the motor acceleration waveforms, namely 4.4 seconds.

2) Switching frequency is shown in table 2.

3) We have not evaluated the power loss and cannot give an exact quantitative value of the quality indicators. We considered only various control algorithms during motor acceleration.

4) The paper presents results of a predominantly qualitative type. In future studies with quantitative results, we will use the summary table.

In addition, together with the translators, we have also changed the wording of the text for greater clarity, thereby improving the level of English.

Thanks for your expert review!

Reviewer 4 Report

In this manuscript, a multi-zone AC voltage converter is presented that can be used as a soft starter for IM. There are some significant concerns that should be addressed by the authors.

1. The authors claimed that use of capacitive voltage dividers will also compensate for the consumed reactive power without any more investigations.

2. The switch equivalent circuit in figure 3 is not clear. Why a diode is added to a traditional bidirectional voltage and current switch? More analysis with convincing explanations is highly needed.

3. Authors claimed that the overall AC voltage regulator efficiency remains not of the very high value, mostly due to the double energy conversion structures. The proposed structure is a single stage converter, but there are three series semiconductors in each switch of this structure, which increases the conduction loss of switches. Efficiency analysis and comparison with traditional soft starters is required from my point of view.

4. The Introduction section has not been written acceptably. It is recommended to introduce the necessity of the research based on challenges of the literature. The novelty and main contributions of the paper should be introduced clearly.

Author Response

Good afternoon!

We express our gratitude to you for the review of our article.

Below are responses to comments.

1) In the description of the 23rd figure, when connecting to a resistive load, we added a description in which we emphasized that the load current is ahead of the voltage, which is explained by the presence of the capacitive structure of the converter. When using an active-inductive load, it is possible to reduce the shift between the load current and voltage, thereby reducing the angle and increasing the power factor, and as a result, compensate for the consumed reactive power.

2) The role of the diode is to prevent a short circuit when opening, for example, 2 zones, so that the second capacitor C2, when 1 zone is already open, does not have huge switching currents. The use of just such a structure of power switches will allow, together with the synchronization unit with the network, to provide a correct control system.

3) The purpose of the article was not a detailed comparison and analysis of power loss calculations and efficiency of various soft starters. The purpose of the article is control algorithms for a multi-zone AC converter for soft start of induction motors. We plan to pay attention to the calculation of efficiency in our future work on comparison and review.

4) The need for research is due to the lack of competitive single-stage soft starters with the ability to limit the starting current and set the desired acceleration curve, without using a combination of rectifier and inverter (frequency converter). The objectives are clearly presented in the second half of the introduction section.

In addition, together with the translators, we have also changed the wording of the text for greater clarity, thereby improving the level of English.

Thanks for your expert review!

Reviewer 5 Report

I have the following observations:

1.     The paper presents a study of the control system 16 of a multi-zone AC voltage regulator.

2.     I suggest to review the Figures axes (must be write the measurement unity and the exanimated quantity)

3.     The paper it’s well written.

4.     Please summarize the performance of  yours algorithms in a table with specifications of quantities in question.

Author Response

Good afternoon!

We express our gratitude to you for the review of our article.

Below are responses to comments.

1) The paper presents a study of control algorithms for a multi-zone AC converter for soft start of induction motors with a limited starting current and setting the desired acceleration curve.

2) Absolutely all oscillograms and graphs have not only signatures, but also an indication of the units of measurement.

4) The purpose of the article was not to compare algorithms in terms of performance. We will certainly take this remark into account in our further studies.

In addition, together with the translators, we have also changed the wording of the text for greater clarity, thereby improving the level of English.

Thanks for your expert review!

Reviewer 6 Report

The paper proposes an multi-zone AC voltage converter for a soft starter of an induction motor. In addition, the paper uses capacitor voltage dividers to compensate for the reactive power. A simulation program of Matlab/Simulink and experimental results are both provided. The paper proposes multi-level voltages  for a three-phase load. The idea is new and interesting. However, there are several points need to be explained as follows:

1. The authors should compare this method and 3-phase inverter. What are the advantages and disadvantages of this method when compared starting the motor with an inverter.

2. In Fig. 17, the current ripples of the motor is large. Could authors provide some comments.

3. It seems the capacitor current ripples are large. The current ripples may reduce the life of capacitors. Could authors provide some comments.

4.  Can this method use for unbalanced three-phase load?

5. Should authors consider V/F  constant method to main the fluxes of motor constant? 

Author Response

Good afternoon!

We express our gratitude to you for the review of our article.

Below are responses to comments.

1 and 5) We have added a comparison of our converter with a combination of rectifier and inverter (frequency converter). The comparison is predominantly qualitative rather than quantitative. We will definitely add comparison of quality indicators in future studies and pay more attention to them.

2) The paper presents a study of control algorithms for a multi-zone AC voltage converter for soft start of induction motors with a limited starting current and setting the desired acceleration curve. The spectral composition of the consumed current has mainly the main harmonics (50 Hz), and all the higher harmonics make an insignificant contribution to the shape, and hence to the ripple range, which is confirmed by the low THD coefficient.

3) We really paid a lot of attention to this problem, but did not cover this material in the article. The calculation of the converter capacitance values is based on the limitation of the current amplitudes by setting the value of the capacitor reactance.

4) The mathematical calculation of the DEA method is applicable for asymmetric three-phase networks. However, such a calculation will require high computing power, since the number of reactive elements will increase, and hence the number of equations in the system.

In addition, together with the translators, we have also changed the wording of the text for greater clarity, thereby improving the level of English.

Thanks for your expert review!

Round 2

Reviewer 2 Report

No more comments.

Reviewer 3 Report

well written with the improved version from the previous. Double check format and English. 

Reviewer 4 Report

Some of my comments have not addressed by the authors. Nevertheless, the paper can be considered for publication from my point of view.